# The roles of online and offline replay in planning

Eran Eldar[1,2,3]*, Gaëlle Lièvre[2,3], Peter Dayan[4,5†], Raymond J Dolan[2,3†]

[1]Departments of Psychology and Cognitive Sciences, Hebrew University of Jerusalem, Jerusalem, Israel; [2]Max Planck UCL Centre for Computational Psychiatry and Ageing Research, University College London, London, United Kingdom; [3]Wellcome Centre for Human Neuroimaging, University College London, London, United Kingdom; [4]Max Planck Institute for Biological Cybernetics, Tübingen, Germany; [5]University of Tübingen, Tübingen, Germany

**Abstract** Animals and humans replay neural patterns encoding trajectories through their environment, both whilst they solve decision-making tasks and during rest. Both on-task and off-task replay are believed to contribute to flexible decision making, though how their relative contributions differ remains unclear. We investigated this question by using magnetoencephalography (MEG) to study human subjects while they performed a decision-making task that was designed to reveal the decision algorithms employed. We characterised subjects in terms of how flexibly each adjusted their choices to changes in temporal, spatial and reward structure. The more flexible a subject, the more they replayed trajectories during task performance, and this replay was coupled with re-planning of the encoded trajectories. The less flexible a subject, the more they replayed previously preferred trajectories during rest periods between task epochs. The data suggest that online and offline replay both participate in planning but support distinct decision strategies.

**\*For correspondence:**
eran.eldar@mail.huji.ac.il

[†]These authors contributed equally to this work

**Competing interests:** The authors declare that no competing interests exist.

## Introduction

Online and offline replay are both suggested to contribute to decision making (*Behrens et al., 2018*; *Diba and Buzsáki, 2007*; *Foster, 2017*; *Foster and Wilson, 2006*; *Gupta et al., 2010*; *Ji and Wilson, 2007*; *Kurth-Nelson et al., 2016*; *Louie and Wilson, 2001*; *Ólafsdóttir et al., 2017*; *Pezzulo et al., 2014*; *Skaggs and McNaughton, 1996*; *Ólafsdóttir et al., 2015*; *Stachenfeld et al., 2017*), but their precise contributions remain unclear. Replay of experienced and expected state transitions during a task, either immediately before choice or following outcome feedback, is particularly well suited to mediate on-the-fly planning, where choices are evaluated based on the states to which they lead (this is known as model-based planning). Off-task replay might serve a complementary role of consolidating a model of a state space, which specifies how each state can be reached from other states as well as the values of each state. According to this perspective, both types of replay help subjects make choices that are flexibly adapted to current circumstances.

However, an alternative possibility is that off-task replay also directly participates in planning, by calculating and storing a (so-called model-free) decision policy that specifies in advance what to do in each state (*Gershman et al., 2014*; *Mattar and Daw, 2018*; *Momennejad et al., 2018*; *Sutton, 1991*). Such a pre-formulated policy is inherently less flexible than a policy that is constructed on the fly, but at the same time it decreases a need for subsequent online planning when time itself might be limited. Thus, rather than online and offline replay both supporting the same form of planning, this latter perspective suggests a trade-off between them. In other words, online replay promotes on-the-fly (model-based) flexibility, whereas offline replay establishes a stable (model-free) policy.

**eLife digest** Studies show that humans and animals replay past experiences in their brain. To do this, the brain creates a pattern of electrical activity for each part of a multistep experience and then plays them back in order. Humans and other animals can replay scenarios either while the experience is still happening (i.e. online replay) or later when they are resting or sleeping (i.e. offline replay). Being able to replay an experience and its outcome may help a person or animal plan a better course of action in the future. However, it is poorly understood how online and offline replay each contribute to such planning.

To answer this question, Eldar et al. used a brain imaging tool called magnetoencephalography (MEG for short) to measure the electrical activity inside the brain. This technique was able to detect replays in the brain of individuals performing a particular task, and later whilst they were resting.

In the experiments, 40 healthy volunteers played a game in which each location in a space was associated with an image, for example a frog or a traffic sign, and each image was given a value. Participants got paid for moving to more valuable images in one or two steps. Eldar et al. found that people who replay their steps during a task are able to adjust their choices on the fly, whereas individuals who replay their choices during rests tend to approach a task with a less flexible, more preformed plan.

Eldar et al. suggest that replaying an experience too much during rest and not enough in real-time might contribute to more rigid behaviors, a theory that could shed light on the mechanisms behind certain behavioral disorders such as obsessive compulsive disorder. However, more studies are needed to determine if these two different replay strategies play a causal role in human behavior.

Despite the wide-ranging behavioural implications of a distinction between model-based and model-free planning (*Crockett, 2013*; *Everitt and Robbins, 2005*; *Gillan et al., 2017*; *Kurdi et al., 2019*), and much theorising on the role of replay in one or the other form of planning, to date there is little data indicating whether online and offline replay have complementary or contrasting impacts in this regard. However, recent advances in magnetoencephalography (MEG) analysis have made it possible to study replay in human subjects in relation to learning and decision-making behaviour (*Eldar et al., 2018*; *Kurth-Nelson et al., 2016*; *Liu et al., 2019*). This methodology involves three key steps. First, MEG signals are recorded before, during, and after the subject per-forms a task of interest. Second, the MEG time series are decoded so as to estimate a moment-by-moment probability a subject is neurally representing each task element, even when the sensory processing associated with that element has ceased. This is the minimal requirement for replay. However, a special feature of replay is the coordination between the non-sensory neural representa-tions of multiple, related, task elements. Thus, finally, the relationships between elements' represen-tational probability time series are examined to determine whether pairs of elements tended to be represented sequentially, one after the other.

Here, we use this methodology to test the relationship between both online and offline replay and key aspects of decision flexibility that dissociate model-free and model-based forms of planning (*Daw et al., 2011*). For this purpose, we first recorded MEG signals from human subjects during rest and while they navigated a specially designed state space. We next characterised each individual subject's decision-making flexibility based on their task behaviour, and we then analysed their MEG signals to look for evidence of on-task (*Eldar et al., 2018*) and off-task (*Kurth-Nelson et al., 2016*; *Liu et al., 2019*) sequences of state representations.

## Results

### Individual differences in decision flexibility

We used distinct visual images to represent eight unique states, where occupancy of each state pro-vided a different amount of reward (*Figure 1a*). Subjects started each trial at a random state and had to choose a movement direction in order to collect reward from subsequent states (*Figure 1b*). Subjects learnt beforehand how much reward was associated with each state (*Figure 1—figure*

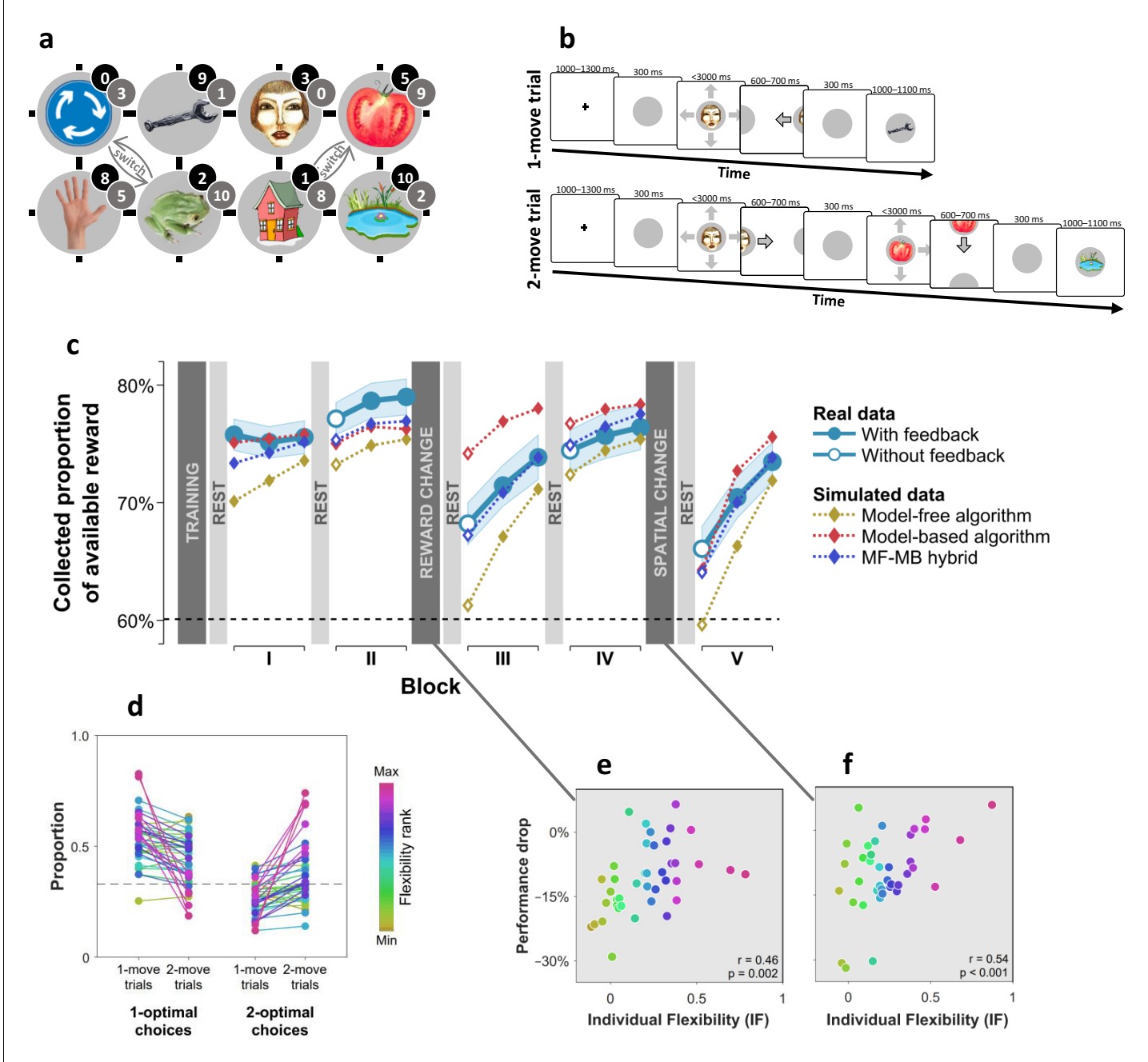

**Figure 1.** Subjects differed in decision flexibility. (**a**) Experimental task space. Before performing the main task, subjects learned state-reward associations (numbers in black circles) and they were then gradually introduced to the state space in a training session. After performing the main task for two blocks of trials, subjects learned new state-reward associations (numbers in dark grey circles) and then returned to the main task. Before a final block of trials, subjects were informed of a structural task change such that 'House' switched position with 'Tomato', and 'Traffic sign' switched position with 'Frog'. The bird's eye view shown in the figure was never seen by subjects. Subjects only saw where they started from on each trial and, after completing a move, the state to which their move led. The map was connected as a torus (e.g., starting from 'Tomato', moving right led to 'Traffic sign', and moving up or down from the tomato led to 'Pond'). (**b**) Each trial started from a pseudorandom location from whence subjects were allowed either one ('1-move trial') or two ('2-move trial') consecutive moves (signalled at the start of each set of six trials), before continuing to the next trial. Outcomes were presented as images alone, and the associated reward points were not shown. A key design feature of the map was that in 5 out of 6 trials the optimal (first) move was different depending on whether the trial allowed for one or two moves. For instance, given the initial image-reward associations (black) and image positions, the best *single* move from 'Face' is LEFT (9 points), but when two moves are allowed the best moves are RIGHT and then DOWN (5+9 giving 15 total points). Note that the optimal moves differed also given the second set of image-reward associations. On 'no-feedback' trials (which started all but the first block), outcome images were also not shown (i.e., in the depicted trials, the 'Wrench', 'Tomato' and

*Figure 1 continued on next page*

*Figure 1 continued*

'Pond' would appear as empty circles). (c) The proportion of obtainable reward points collected by the experimental subjects, and by three simulated learning algorithms. Each data point corresponds to 18 trials (six 1-move and twelve 2-move trials), with 54 trials per block. The images to which subjects moved were not shown to subjects for the first 12 trials of Blocks II to V (the corresponding 'Without feedback' data points also include data from 6 initial trials with feedback wherein starting locations had not yet repeated, and thus, subjects' choices still reflected little new information). All algorithms were allowed to forget information so as to account for post-change performance drops as best fitted subjects' choices (see Materials and methods for details). Black dashed line: chance performance. Shaded area: SEM. (d) Proportion of first choices that would have allowed collecting maximal reward where one ('1-optimal') or two ('2-optimal') consecutive moves were allowed. Choices are shown separately for what were in actuality 1-move and 2-move trials. Subjects are colour coded from lowest (gold) to highest (red) degree of flexibility in adjusting to one vs. two moves (see text). Dashed line: chance performance (33%, since up and down choices always lead to the same outcome). (e,f) Decrease in collected reward following a reward-contingency (e) and spatial (f) change, as a function of the index of flexibility (IF) computed from panel d. Measures are corrected for the impact of pre-change performance level using linear regression. *p* value derived using a premutation test.

The online version of this article includes the following figure supplement(s) for figure 1:

**Figure supplement 1.** Image-reward training: Timeline of a trial.
**Figure supplement 2.** Example sketches of the state space by a representative subject.
**Figure supplement 3.** Evidence of advance prospective planning in flexible subjects.
**Figure supplement 4.** Individual flexibility reflected the balance between MB and MF planning.

*supplement 1*), but they did not know initially where states were in relation to one another. The latter aspect of task structure had to be acquired through trial and error learning in order for subjects to be able to implement subsequent moves that delivered the maximal amount of reward. We assessed subjects' flexibility in three ways. First, after the initial two blocks of trials, we changed the reward associated with each state (*Figure 1a*; grey numbers) such that persisting with previously optimal moves would result in below-chance performance. Second, after two additional blocks of trials, we informed subjects that two specified pairs of states had switched positions (*Figure 1a*; 'switch'), again rendering the previously optimal policy now suboptimal. A flexible model-based planner would be capable of re-planning their moves perfectly following each of these instructed changes, since such a planner has acquired knowledge as to how each state can be reached. Conversely, a pure model-free planner would need to learn a new policy from scratch via trial and error each time there is a change, since such an agent only possesses a now outdated policy that specifies where to move from each state.

Examining how subjects' overall performance altered immediately following these changes revealed a decrement in average performance (*Figure 1c*). However, there were substantial individual differences in this regard, with some subjects seamlessly adapting to reward and position changes, and others showing drops in performance to chance levels. Subjects whose performance showed a strong decline following a reward change tended to cope poorly also with the position change ($\rho = 0.50$, partial correlation controlling for performance levels before the changes; $p = 0.001$, Permutation test).

As a third, more continuous, test of a different aspect of decision flexibility, we interleaved sets of six trials in which only a single move was allowed ('1-move trials') with trials which allowed two consecutive moves ('2-move trials'; *Figure 1b*). In 2-move trials, subjects were rewarded for both states they visited, and thus, an optimal course of action often required subjects to move first to an initial low-reward state in order to gain access to a high-reward state with their second move. Thus, we defined an individual index (IF) of decision flexibility as the difference between the proportion of moves that were optimal given the actual number of allotted moves and the proportion of moves that would have been optimal given a different number of allotted moves (i.e., had 1-move trials instead involved two moves and 2-move trials involved one move). An IF value of zero implies no net adjustment, while positive IF values imply advantageous flexibility.

The results indicate subjects adjusted their choices advantageously to the number of allotted moves (+0.21, SEM 0.05, p < 0.001, Bootstrap test), though there was evidence again of substantial individual differences (*Figure 1d*). Importantly, IF correlated with how well a subject coped with the reward-contingency (*Figure 1e*) and position (*Figure 1f*) changes as well with how accurately they could sketch maps of the state space at the end of the experiment ($r = 0.51$, *p<0.001*, Permutation test; *Figure 1—figure supplement 2*), indicating these subjects acquired and utilised a model of the state space.

## Planning two steps into the future

Having a cognitive model that specifies how states are spatially organised makes it possible to plan several steps into the future. To test whether subjects were able to do that, we challenged subjects with 12 'without-feedback' trials at the beginning of each of the last four blocks, during which outcome images were not shown. This meant that in 2-move trials subjects had to choose their second move 'blindly', without having seen the image to which their previous move had led (e.g., the tomato in *Figure 1b*). We found that subjects performed above chance on these blind second moves (proportion of optimal choices: 0.56, SEM 0.03; chance = 0.45; p<0.001, Bootstrap test), and this was the case even immediately following position and reward changes, when subjects could not have relied on previously tested 2-move sequences (0.52, SEM 0.03; p=0.01, Bootstrap test). Most importantly, such blind-move success was correlated with IF (*Figure 1—figure supplement 3a*).

This result indicates that more flexible subjects were better able to plan two steps into the future when required. Examining response times suggested flexibility was associated with advance planning also when it was not required. Thus, we found that IF correlated with quicker execution of second moves in general (Spearman correlation with median reaction time: $r = -0.61$, $p<0.001$, Permutation test). To determine whether advance planning was indeed generally associated with flexibility, we examined at what point during a trial their choices became decodeable from MEG signals. For this purpose, we trained a decoder to decode chosen moves from MEG signals recorded outside of the main task (see Materials and methods for details). Validating the decoder on MEG data from the main task showed that chosen moves became gradually more evident over the course of the trial, their decodability peaking 140 ms before a choice was made (*Figure 1—figure supplement 3b*).

Thus, we used the move decoder to test whether second-move choices began to materialise in the MEG signal even before subjects observed the outcomes of their first moves. We found that chosen second moves were indeed decodeable already during first-move choices (decodability: $M = 0.006$, 95% Credible Interval = 0.004 to 0.008, Bayesian Gaussian Process analysis; *Figure 1—figure supplement 3c*) and prior to the appearance of the first outcome (decodability: $M = 0.004$, 95% Credible Interval = 0.002 to 0.006; *Figure 1—figure supplement 3d*). Importantly, this early decodability was correlated with IF ($\beta$: $M = 0.29$, 95% Credible Interval = 0.24 to 0.34). By contrast, later decodability, following the onset of the second image, did not correlate with IF ($\beta$: $M = 0.02$, 95% Credible Interval = $-0.02$ to 0.05). Thus, neural and behavioural evidence concur with the notion that flexibility was associated with planning second moves prospectively.

## Individual flexibility reflected MF-MB balance

These convergent results suggest that IF reflected deployment of a model-based planning strategy. To test this formally, we compared how well different model-free (MF) and model-based (MB) decision algorithms, as well as a combination of both, explained subjects' choices. Importantly, we enhanced these algorithms to maximise their ability to mimic one another (see Materials and methods for details). Thus, for instance, the MF algorithm included separate 1-move and 2-move policies, which allow it to achieve optimal adjustment to trial type given sufficient experience.

We found that a hybrid of MF and MB algorithms outperformed substantially either of them alone (Bayesian Information Criterion [*Bishop, 2006*]: MF = 40821, MB = 43249, MF-MB hybrid = 39908), suggesting that subjects employed a mix of model-free and model-based planning strategies. Simulating task performance using the hybrid algorithm showed it captured adequately behavioural differences evident between subjects (correlation between real and simulated IF: $r = 0.92$, $p<0.001$, Permutation test; *Figure 1—figure supplement 4a*). When we examined each subject's best-fitting parameter values, to determine which of these covaried with IF, we found 84% of inter-individual variance in IF was explained by three parameters that control a balance between model-based and model-free planning (*Figure 1—figure supplement 4b*). Importantly, less flexible subjects had comparable learning rates and a higher model-free inverse temperature parameter (in 2-move trials), indicating that lower flexibility did not reflect a non-specific impairment, but rather, it was associated with enhanced deployment of a model-free algorithm. Thus, our index of flexibility specifically reflected the influence of model-based, as compared to model-free, planning.

## On-task replay is induced by prediction errors and associated with high flexibility

In rodents, reinstatement of past states, potentially in the service of planning, is evident both prior to choices (*Pfeiffer and Foster, 2013*) and following observation of outcomes (*Pezzulo et al., 2014*). Thus, we determined firstly at what point states were neurally reinstated during our task. For this purpose, we trained MEG decoders to identify the images subjects were processing (*Figure 2a*). Such decoders robustly reveal stimulus representations that are reinstated from memory and contribute to decision processes (*Eldar et al., 2018*; *Kurth-Nelson et al., 2015*; *Bishop, 2006*; *Pfeiffer and Foster, 2013*). Crucially, image decoders were trained on MEG data collected prior to subjects having any knowledge about the task, ensuring that the decoding was free of confounds related to other task variables (*Figure 2—figure supplement 1*). Applying these decoders to MEG signals from the main task, we found no evidence of prospective representation of outcome states (images) to which subjects will transition at choice (*Figure 2—figure supplement 2a*). Instead, we found strong evidence that following outcomes (corresponding to new states to which subjects had transitioned), subjects represented the states from which they had just moved ($\bar{t} = 3.4$, $p = 0.001$, Permutation test; *Figure 2—figure supplement 2b*). Consequently, we examined in detail the MEG data recorded following each outcome for evidence of replay of state sequences that subjects had just traversed.

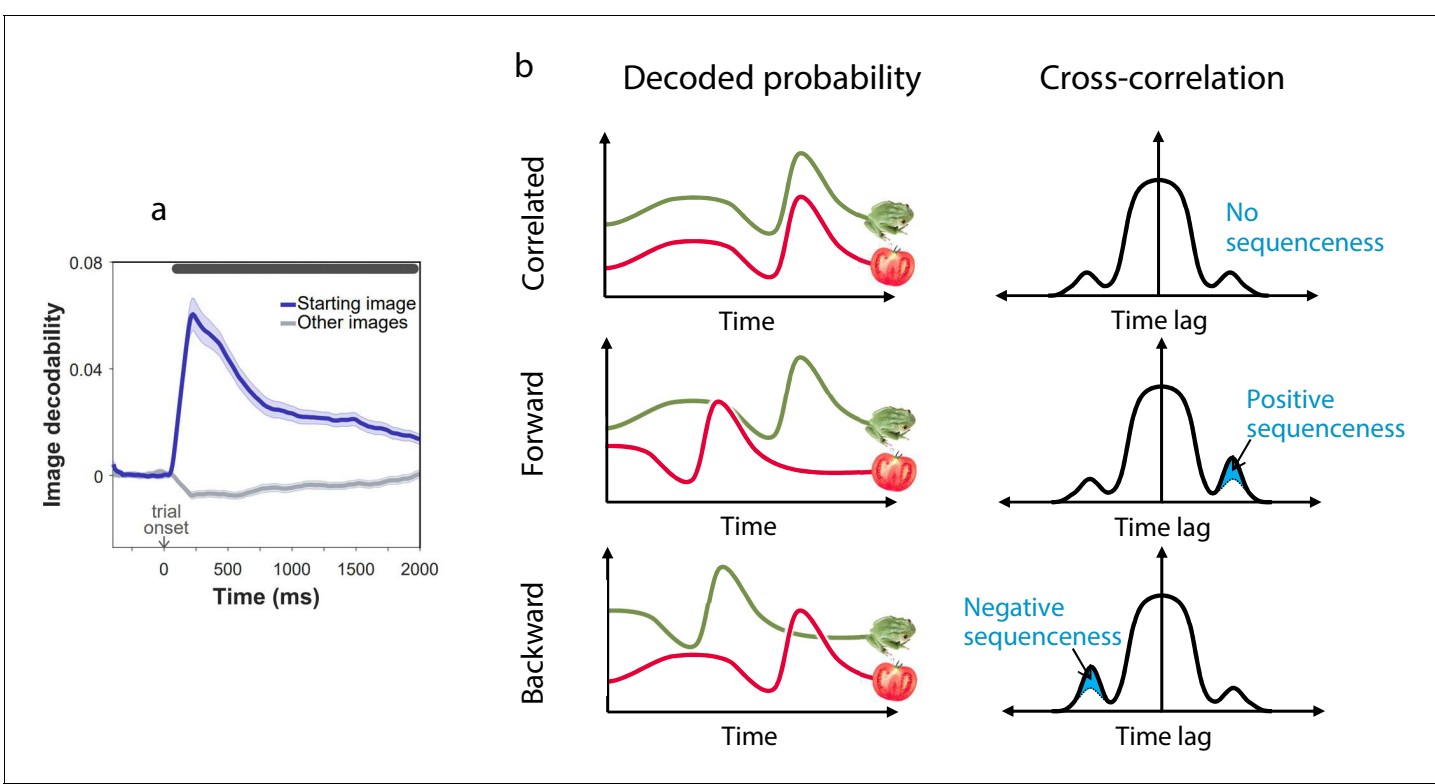

**Figure 2.** Sequenceness analysis. (a) Validation of the image MEG decoder used for the sequenceness analyses. *n* = 40 subjects. The plot shows the decodability of starting images from MEG data recorded during the main task at trial onset. Decodability was computed as the probability assigned to the starting image by an 8-way classifier based on each timepoint's spatial MEG pattern, minus chance probability (0.125). (b) Schematic depiction of the sequenceness analysis used to determine whether representational probabilities of pairs of elements followed one another in time. Sequenceness is computed as the difference between the cross-correlation of two time series with positive and negative time lags. Since it focuses on asymmetries in the cross correlation function, this measure is useful for detecting sequential relationships even between closely correlated (or anti-correlated) time series. Negative sequenceness indicates signals are ordered in reverse.

The online version of this article includes the following figure supplement(s) for figure 2:

**Figure supplement 1.** Decoding procedure.
**Figure supplement 2.** Previous, not subsequent, states were encoded in MEG.

To test for evidence of replay, we applied a measure of 'sequenceness' to the decoded MEG time series, a metric we previously showed is sensitive in detecting replay of experienced and decision-related sequences of states (*Eldar et al., 2018*; *Kurth-Nelson et al., 2016*; *Liu et al., 2019*; *Figure 2b*). Importantly, sequenceness is not sensitive to simultaneous covariation, and thus, it is only found if stimulus representations follow one another in time (*Eldar et al., 2018*; as in previous work, we allowed for inter-stimulus lags of up to 200 ms). Thus, following each outcome, we computed sequenceness between the decoded representations of the preceding and the outcome state (*Figure 3a*). Additionally, MEG signals recorded following the second outcome in 2-move trials were also tested for sequenceness reflecting the trial's first transition (i.e., between the starting state and first outcome; *Figure 3b*).

Using an hierarchical Bayesian Gaussian Process approach (see Materials and methods for details) we tested for timepoints at which sequenceness was evident and correlated with individual flexibility. This method directly corrects for comparison across multiple timepoints by accounting for the dependency between them (*Kruschke, 2014*). Since replay is thought to be induced by surprising observations (*Mattar and Daw, 2018*; *Momennejad et al., 2018*; *Moore and Atkeson, 1993*; *Peng, 1993*), we also included surprise about the outcome (i.e. the state prediction error inferred by the hybrid algorithm) as a predictor of sequenceness. We found significant sequenceness encoding the last experienced state transition (from 50 to 330 ms and from 820 to 950 ms following outcome onset; *Figure 3a*; note that the median split is for display purposes alone; actual analyses depended on the continuous flexibility index) and, at the conclusion of 2-move trials, also the penultimate transition (from 130 ms before to 350 ms following outcome onset; *Figure 3b*). These sequences were accelerated in time, with an estimated lag of 130 ms between the images, and were encoded in a 'forward' direction corresponding to the order actually visited. Moreover, later in the post-outcome epoch, the penultimate transition was also replayed backwards (from 440 to 940 ms following outcome onset). As would be expected, the finding of sequenceness was associated with enhanced decoding of previously visited states during the post-outcome epoch (*Figure 2—figure supplement 2c*).

More importantly, we found this evidence of replay, across all timepoints, was correlated with IF (mean $\beta = 0.17$, 95% Credible Interval = 0.13 to 0.20), with surprise about the outcome (mean $\beta = 0.06$, CI = 0.03 to 0.10), and with the interaction of these two factors (mean $\beta = 0.19$, CI = 0.15 to 0.22). Thus, sequenceness was predominantly evident following surprising outcomes in subjects with high index of flexibility. This result is consistent with online replay contributing to post-outcome model-based planning.

## On-task replay is associated with changes of policy

Recent theorising regarding the role of replay in planning argues that replay is preferentially induced when there is benefit to updating one's policy (*Mattar and Daw, 2018*). Although policy updates can either reinforce or inhibit a chosen move, in our experiment, subsequently avoided choices in our experiment were more likely followed by a policy update than subsequently repeated choices, since a substantial proportion of the latter already reflected a well-informed policy (as evidenced by subjects' above chance performance in *Figure 1c*). Thus, a role for replay in planning predicts that subjects should be more disposed to replay trajectories that they might not want to choose again, rather than trajectories whose choice reflects a firm policy. To determine whether decodable on-task replay was associated with behavioural change, we tested the relationship between sequenceness corresponding to each move that subjects chose, and the probability of making a different choice when occupying the same state later on. We found that moves after which high forward sequenceness was evident corresponded to moves that were less likely to be re-chosen subsequently (*Figure 3c*). These policy changes increased the proportion of obtained reward ($M = +11.1\%$, $SEM = 1.5\%$, $p = 0.001$). Thus, we found evidence of online replay for a chosen trajectory was coupled with advantageous re-evaluation of one's choice.

## Off-task replay is induced by prediction errors and associated with low flexibility

We next studied off-task replay, examining MEG data recorded during the 2-minute rest period that preceded each experimental block. Since each block included five frequently repeating starting

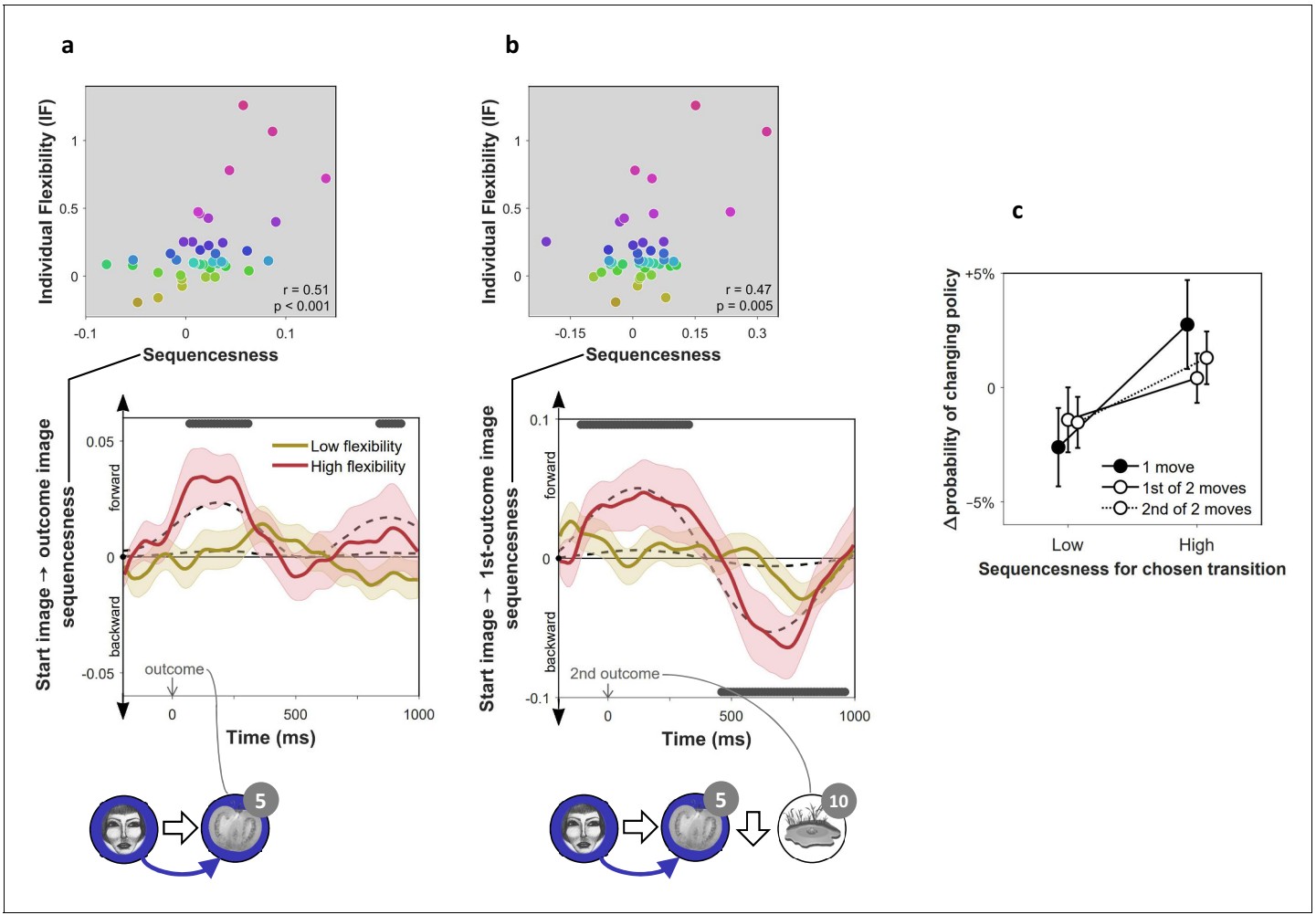

**Figure 3.** On-task replay of state-to-state trajectories as a function of individual flexibility. $n$ = 40 subjects. (a) Sequenceness corresponding to a transition from the image the subject had just left ('Start image'; in the cartoon at the bottom, the face) to the image to which they arrived ('outcome image'; the tomato) following highly surprising outcomes (i.e., above-mean state prediction error). In the cartoon, the white arrow indicates the actual action taken on the trial; the blue arrow indicates the sequence that is being decoded. For display purposes alone, mean time series are shown separately for subjects with high (above median) and low (below median) IF. Positive sequenceness values indicate forward replay and negative values indicate backward replay. As in previous work (*Eldar et al., 2018*), sequenceness was averaged over all inter-image time lags from 10 ms to 200 ms, and each timepoint reflects a moving time window of 600 ms centred at the given time (e.g., the 1 s timepoint reflects MEG data from 0.7 s to 1.3 s following outcome). Dashed lines show mean data generated by a Bayesian Gaussian Process analysis, and the dark gray bars indicate timepoints where the 95% Credible Interval excludes zero and Cohen's $d > 0.1$. The top plot shows IF as a function of sequenceness for the timepoint where the average over all subjects was maximal. $p$ value derived using a premutation test. Dot colours denote flexibility rank. (b) Sequenceness following the conclusion of 2-move trials corresponding to a transition from the starting image to the first outcome image. (c) Difference in the probability of subsequently choosing a different transition as a function of sequenceness recorded at the transition's conclusion. For display purposes only, sequenceness is divided into high (i.e., above mean) and low (i.e., below mean). A correlation analysis between sequenceness and probability of policy change showed a similar relationship (Spearman correlation: $M = -0.04$, $SEM = 0.02$, $p = 0.04$, Bootstrap test). Sequenceness was averaged over the first cluster of significant timepoints from panels b) and c), in subjects with non-negligible inferred sequenceness (more than the standard deviation divided by 10; $n = 25$), for the first time the subject chose each trajectory. Probability of changing policy was computed as the frequency of choosing a different move when occupying precisely the same state again. 0 corresponds to the average probability of change (51%). Error bars: s.e.m.

states, we computed sequenceness for the five most frequent image-to-image transitions subjects chose before and after each rest period (mean choice frequency = 8.4 repetitions per block). As a control analysis, we also examined sequenceness for the five least frequently chosen transitions from the same starting states (mean choice frequency = 1.0 repetitions per block). We found significant evidence for sequenceness throughout the rest periods for frequent transitions ($M = 0.002$, $SEM = 0.001$, $p = 0.01$, Bootstrap test). By contrast, we found no evidence of sequenceness for the

infrequent transitions ($M < 0.001$, $SEM = 0.001$, $p = 0.47$, Bootstrap test). Frequent transitions were replayed in a forward direction, with an estimated time lag of 180 ms between images, and prioritised trajectories that induced more reward prediction errors in the previous block (correlation of sequenceness with sum of absolute model-free reward prediction errors inferred by the hybrid algorithm: $M = 0.04$, $SEM = 0.018$, $p = 0.03$, Bootstrap test). Most importantly, off-task sequenceness correlated negatively with IF (*Figure 4*). This association of sequenceness during rest with low flexibility is consistent with a proposed role for offline replay in establishing model-free policies (*Gershman et al., 2014*; *Mattar and Daw, 2018*; *Momennejad et al., 2018*; *Sutton, 1991*).

## Off-task replay can predict subsequently chosen sequences

If offline replay is involved in planning, then its content should predict subjects' subsequent choices. To test this, we dissociated the replay of experienced trajectories from that of planned trajectories, by focusing on the third rest period after which the optimal image-to-image transitions changed entirely (due to a change in state-reward associations; see *Figure 1*). As subjects had been taught about the reward change before this rest period, the rest afforded subjects an opportunity to re-plan their choices accordingly.

We first examined the behavioural effect of the state-reward change in more detail. The most frequently chosen transitions in the block that followed the third rest period differed from the transitions most frequently chosen in the preceding block (overlap: $M = 14\%$, $SEM = 3\%$), and this policy change was substantially greater than for the other rest periods (overlap: $M = 53\%$, $SEM = 2\%$). As expected, the newly chosen transitions from the following block were advantageous given the new state-reward associations (reward collected: $M = 71\%$, $SEM = 2\%$; chance = 60%) and disadvantageous given the state-reward associations that had so far applied ($M = 52\%$, $SEM = 2\%$).

Given the behavioural change, we focused our examination of the MEG data on evidence for sequenceness during this crucial third rest period. We found that subjects indeed replayed the transitions they subsequently chose ($M = 0.004$, $SEM = 0.002$, $p = 0.02$, Bootstrap test). This replay of subsequently chosen moves indicates subjects utilised a model of the task to re-plan their moves offline (*Momennejad et al., 2018*; *Gershman et al., 2014*; *Mattar and Daw, 2018*). Our reasoning here is that re-planning in light of the new reward associations, before subjects experienced them in practice, requires a model that specifies how to navigate from one state to another. Indeed, multiple regression analysis showed that low IF was only associated with sequenceness encoding previously chosen transitions ($\beta = -0.35$, $t_{37} = 2.25$, $p = 0.03$), whereas the replay of subsequently chosen transitions did not correlate with IF ($\beta = -0.004$, $t_{37} = 0.03$, $p = 0.97$). On the other hand, the lack of a flexibility enhancement associated with prospective offline replay might indicate that, as might be expected, offline planning is ill-suited for enhancing trial-to-trial flexibility.

## Discussion

We find substantial differences in the behaviour of individual subjects in a simple state-based sequential decision-making task that corresponds also to a distinction in the nature, and apparent effects, of MEG-recorded on- and off-task replay of state trajectories (*Figure 5*). These results bolster important behavioural dissociations, as well as provide substantial new insights into the control algorithms that subjects employ. The findings fit comfortably with an evolving literature that addresses human replay and preplay (*Eldar et al., 2018*; *Kurth-Nelson et al., 2015*; *Kurth-Nelson et al., 2016*; *Liu et al., 2019*; *Schuck and Niv, 2019*).

The distinction between model-based and model-free reasoning has intuitive appeal, and close associations with many well-established psychological distinctions (*Kahneman, 2011*; *Stanovich and West, 2000*). However, popular tasks for investigating this distinction (*Daw et al., 2011*; *Decker et al., 2016*; *Gillan et al., 2017*; *Gläscher et al., 2010*) have been criticised for offering a better grasp on model-based compared to model-free reasoning processes (*Gillan et al., 2015*; *da Silva and Hare, 2019*); for rewarding model-based reasoning indifferently (*Kool et al., 2016*); and for admitting complex model-free strategies that can masquerade as being model-based (*Akam et al., 2017*). In our new task, we show a convergence between superficially divergent methods for distinguishing model-based and model-free methods – flexibility to immediate task demands (one-step versus two-step control), preserved performance in the face of changes in the location of rewards or structure, and an ability to reproduce explicitly, after the fact, the transition

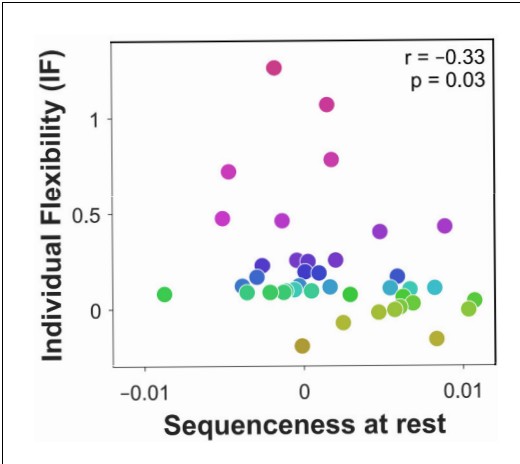

**Figure 4.** Off-task replay of past and future trajectories. *n* = 40 subjects. Individual flexibility as a function of sequenceness in rest MEG data for the five most frequently experienced image-to-image transitions. For each rest period, sequenceness was averaged over transitions from both the preceding and following blocks of trials. *p* value derived using a premutation test. Dot colours denote flexibility rank.

structure. Furthermore, the task effectively incentivises flexible model-based reasoning, as this type of reasoning alone allows collection of substantial additional reward (93%) compared to our most successful MF algorithm (80%). These convergent observations suggest that the model-based and model-free distinction we infer from our task rests on solid behavioural grounds.

In human subjects, there is a growing number of observations of replay and/or preplay of potential trajectories of states that are associated with the structure of tasks that subjects are performing (*Kurth-Nelson et al., 2015*; *Kurth-Nelson et al., 2016*; *Schuck and Niv, 2019*). However, it has been relatively hard to relate these replay events to ongoing performance. By contrast, there is evidence that rodent preplay has at least some immediate behavioural function (*Gupta et al., 2010*; *Pfeiffer and Foster, 2013*), and there are elegant theories for how replay should be optimally sequenced and structured in the service of planning. In particular, a recent normative model of replay, which aims to account for both online and offline events, suggests forward replay should prioritise trajectories on which the agent might soon re-embark,

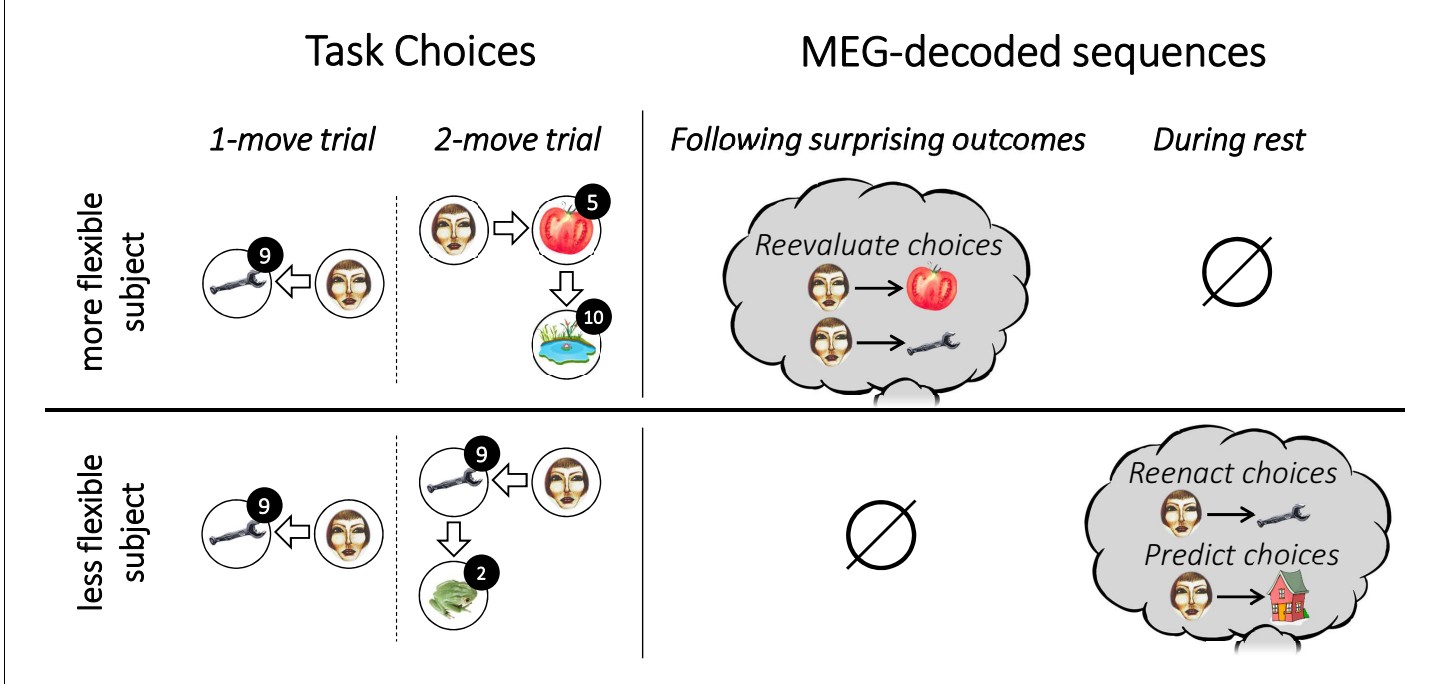

**Figure 5.** Individual flexibility and evidence of replay. The figure illustrates typical results for individuals with low (i.e., below median) and high (i.e., above median) flexibility. More flexible subjects advantageously adjusted their choices to the number of allotted moves. Such flexibility was associated with evidence of primarily forward replay following surprising outcomes, encoding the chosen transitions that led to those outcomes, and coupled with a reevaluation of those choices. Less flexible subjects showed evidence of forward replay during rest, encoding previously and subsequently chosen transitions.

and backward replay should prioritise trajectories for which one's policy can be improved (*Mattar and Daw, 2018*). Our results are broadly consistent with this normative perspective, showing that evidence of forward replay prioritises frequently chosen trajectories, whereas evidence of forward and backward replay follows new observations that can inform one's policy, and indeed predicts appropriate changes in policy.

Two main aspects of our results deviate from this normative perspective. First, we primarily find forward sequenceness following outcomes. Second, rather than preplay immediately prior to choice, we found evidence of on-task replay following feedback alone. One possible explanation of the latter result is that replay before choice is more variable in speed and direction, which would make it more difficult to detect. However, this result might also indicate that upon observing an outcome, subjects immediately decided what moves to make next time they start from the same state. Such a strategy could be computationally expedient in that it minimises the need for retrieval and computation later on, during choice time, when a subject might be under time pressure. This suggests a third potential factor impacting on the timing and content of replay – the need to minimise memory load by embedding new information in ones' policies as soon as it is received.

Critically, the timing and content of replay differed across individuals in a manner that links with their dominant mode of planning. More model-based subjects tended to replay trajectories during learning, predominantly reflecting choices they were likely to reconsider. There have been reports of preferential replay of deprecated trajectories in rodents (*Gupta et al., 2010*; *Carey et al., 2019*). However, those studies are consistent with a more general function for replay (e.g., maintaining the integrity of a map given a biased experience), whereas in our case, replay was closely related to future behaviour.

By contrast, the decodeable replay of more model-free subjects reinstated previously experienced trajectories during rest periods, when DYNA-like mechanisms (*Sutton, 1991*) are hypothesised to compile information about the environment to create an effective model-free policy. This replay of state-to-state transitions suggests that despite a general inability at the end of the task to draw a map accurately, model-free subjects do have implicit access to some form of model, though likely an incomplete one. In any case, the lack of association here between offline replay and ultimate winnings indicates that generating a policy offline might not be a good strategy for a task that requires trial-to-trial flexibility.

This aspect of the task might explain an apparent discrepancy with previous work on retrospective revaluation (*Gershman et al., 2014*; *Momennejad et al., 2018*), which indicates offline replay is associated with a greater degree of behavioural change. In the retrospective revaluation paradigm, behavioural change manifests between experimental phases that are separated in time by several minutes. The only way an algorithm like Dyna would be able to afford the trial-to-trial flexibility required in our task would be to furnish different stimulus-response mappings that can be summoned at will. It is possible such flexibility goes beyond the capabilities of any forms of DYNA that might be implemented in the brain.

Our work has a number of limitations. First, although our experimental design probes various facets of decision flexibility, it tests flexibility most extensively by interleaving 1- and 2-move trials. Modelling subjects' choices shows this measure of flexibility captures individual differences in model-based and model-free planning. However, it is important to keep in mind that this measure does not capture all types of decision flexibility, as exemplified, for instance, by the different sort of flexibility that manifests in a retrospective revaluation paradigm. Second, our experiment was not ideally suited for inducing compound representations that link states with those that succeed them, since succession here changed frequently both within and between blocks. However, algorithms that utilise such representations mimic both model-free and model-based behaviour, and future work could utilise our methods to investigate whether and how these algorithms are aided by online and offline forms of replay (*Russek et al., 2017*). Third, the sequenceness measure that we use to determine replay suffers from a restriction of comparing forwards to backwards sequences. There is every reason to expect both forwards and backwards sequences co-exist, so focusing on a relative predominance of one or the other is likely to provide an incomplete picture. The problem measuring forwards and backwards replay against an absolute standard is the large autocorrelation in the neural decoding, and better ways of correcting for this are desirable in future studies. Nevertheless, despite these shortcomings the work we report is a further step towards revealing the rich and divergent structure of human choice in sequential decision making tasks.

## Materials and methods

### Subjects

40 human subjects, aged 18–33 years, 25 female, were recruited from a subject pool at University College London. Exclusion criteria included age (younger than 18 or older than 35), neurological or psychiatric illness, and current psychoactive drug use. To allow sufficient statistical power for comparisons between subjects, we set the sample size to roughly double that used in recent magnetoencephalography (MEG) studies on dynamics of neural representations (*Hunt et al., 2012*; *Kurth-Nelson et al., 2015*), and in line with our previous study of individual differences using similar measurements (including 'sequenceness'; *Eldar et al., 2018*). Subjects received monetary compensation for their time (£20) in addition to a bonus (between £10 and £20) reflecting how many reward points subjects earned in the experiment task. The experimental protocol was approved by the University of College London local research ethics committee, and informed consent was obtained from all subjects.

### Experimental design

To study flexibility in decision making, we designed a 2 × 4 state space where each location was identified by a unique image. Each image was associated with a known number of reward points, ranging between 0 and 10. Subjects' goal was to collect as much reward as possible by moving to images associated with a high numbers of points. Subjects were never shown the whole structure of the state space, and thus, had to learn by trial and error which moves lead to higher reward.

Subjects were first told explicitly how many reward points were associated with each of the eight images. Subjects were then trained on these image-reward associations until they reliably chose the more rewarding image of any presented pair (see Image-reward training).

Next, the rules of the state-space task were explained (see State-space task), and multiple-choice questions were used to ensure that subjects understood these instructions. To facilitate learning, subjects were then gradually introduced to the state space, and were allowed one move at a time from a limited set of starting locations (see State-space training). Following this initial exposure, the rules governing two-move trials were explained and subjects completed a series of exercises testing their understanding of a distinction between one-move and two-move trials (see State-space exercise). Once these exercises were successfully completed, subjects played two full blocks of trials in the state pace, that included both one-move and two-move trials.

We next tested how subjects adapted to a change in the rewards associated with images. For this purpose, we instructed and trained subjects on new image-reward associations (see State-space design). Subjects then played two additional state-space blocks with these modified rewards.

Finally, we tested how subjects adapted to changes in the spatial structure of the state space. For this purpose, we told subjects that two pairs of images would switch locations, informing them precisely which images these were (see State-space design). Multiple-choice questions were used to ensure that subjects understood these instructions. Subjects then played a final state-space block with this modified spatial map.

At the end of the experiment, we also tested subjects' explicit knowledge, asking them to sketch maps of the state spaces and indicate how many points each image was associated with before, and after, the reward contingency changed.

### Stimuli

To ensure robust decoding from MEG, we used eight images that differed in colour, shape, texture and semantic category (*Hunt et al., 2012*; *Carlson et al., 2013*; *Cichy et al., 2014*). These included: a frog, a face, a traffic sign, a tomato, a hand, a house, a pond, and a wrench.

### State-space task

Subjects started each trial in a pseudorandom state, identified only by its associated image. Subjects then chose whether to move right, left, up, or down, and the chosen move was implemented on the screen, revealing the new state (i.e., as its associated image) to which the move led. In 'one-move' trials, this marked the end of the trial, and was followed by a short inter-trial interval. The next trial then started from another pseudorandom location. In 'two-move' trials, subjects made an additional

move from the location where their first move had led. This second move disallowed backtracking the first move (e.g., moving right and then left). Subjects were informed they would be awarded points associated with any image to which they move. Thus, subjects won points associated with a single image on one-move trials, and the combined value of the two images on two-move trials. The numbers of points awarded were never displayed during the main task. Every six trials, short text messages informed subjects what proportion of obtainable reward they had collected in the last six trials (message duration 2500 ms).

Each state-space block consisted of 54 trials, 18 one-move and 36 two-move trials respectively. The first six trials were one-move, the next 12 were two-move trials, then the next six were again one-move trials, the next 12 two-move, and so on. Every six trials, short text messages informed subjects whether the next six trials were going to be one-move or two-move trials (message duration 2000 ms). Every six consecutive trials featured six different starting locations. The one exception to this were the first of the 24 two-move trials of the experiment, where in order to facilitate learning, each starting location repeated for two consecutive trials (a similar measure was also implemented for one-move trials during training; see State-space training). Subjects' performance improved substantially in the second of such pairs of trials (Δproportion of optimal first choices = +0.15, 95% CI = +0.11 to +0.18, p<0.001, Bootstrap test).

At the beginning of every block (except the first one), we tested how well subjects could do the task without additional information, based solely on the identity of the starting locations. For this purpose, images to which subjects' moves led were not shown for the first 12 trials. In two-move trials, this meant subjects implemented a second move from an unrevealed image (i.e., state).

## State-space design

The mapping of individual images to locations and rewards was randomly determined for each subject, but rewards were spatially organised in a similar manner for all subjects. To test whether subjects could flexibly adjust their choices, the state space was constructed such that there were five locations from which the optimal initial move was different depending on whether one or two moves were allowed. We tested subjects predominantly on these starting locations, using all five of them in every six consecutive trials. Following two blocks, the rewards associated with each image were changed, such that the optimal first moves in both 1-move and 2-move trials, given the new reward associations, were different from the optimal moves under the initial reward associations. The initial and modified reward associations were weakly anti-correlated across images ($r = -0.37$). Finally, before the last block, we switched the locations of two pairs of images, such that the optimal first move changed for 15 out of 16 trial types (1- and 2-move trials x 8 starting locations).

## State-space training

Subjects played six short training blocks, each block consisted of 12 one-move trials starting in one of two possible locations. If a subject failed to collect 70% of the points available in one of these short blocks, the block was repeated. The majority of subjects (35 out of 40) had to repeat the first block, whereas only 12% of the remaining blocks were repeated (mean 0.6 blocks per subject, range 0 to 2). Very rarely, a block had to be repeated twice (a total of 5 out of 240 blocks for the whole group). Lastly, subjects played a final training block consisting 48 one-move trials starting at any of the eight possible locations. To facilitate learning, during the first half of the block, each starting location was repeated for two consecutive trials. In the second half of the block, starting locations were fully interleaved.

## State-space exercise

Following the state-space training, which only included one-move trials, we ensured subjects understood how choices should differ in one- and two-move trials by asking them to choose the optimal moves in a series of random, fully visible state spaces. Subjects were given a bird's eye view of each state space, with each location showing the number of reward points with which it was associated. The starting location was indicated in addition to whether one, or two, moves were available from which to collect reward. In all exercises, the optimal initial move was different depending on whether one or two moves were allowed. Every 10 consecutive exercises consisted of 5 one-move trials and five two-move trials. To illustrate the continuity of the state space, the exercise included one-move

and two-move trials, wherein the optimal move required the subject to move off the map and arrive at the other end (e.g., moving left from a leftmost location to arrive at the rightmost location). In another two-move trial, the optimal moves involved moving twice up or twice down, thereby returning to the starting location. Subjects continued to do the exercises until fulfilling a performance criterion of 9 correct answers in 10 consecutive exercises. This criterion was relaxed to eight correct answers if at least 60 exercises had been completed. Only one subject required 60 exercises to reach criterion (mean required exercises = 24.5 exercises, SD 9.3).

### Image-reward training

To ensure subjects remembered how many points each image awarded, we required subjects to select the more rewarding image out of any pair of presented images. First, subjects were asked to memorise the number of points each image would awards. Then, each round of training consisted of 28 trials, testing subjects on all 28 possible pairs of images (*Figure 1—figure supplement 1*). Each trial started with the presentation of one image, depicted on an arrow pointing either right, left, up or down. 800 ms later, another image appeared on an arrow pointing in a different direction. Subjects had then to press the button corresponding to the direction of the more rewarding image. Here, as throughout the experiment, subjects were instructed to press the 'left' and 'up' buttons with their left hand, and the 'right' and 'down' buttons with their right hand. During training, images were mapped to directions such that each of the four directions was equally associated with low- and high-reward images. Once subjects made their choice, the number of points associated with each of the two images appeared on the screen, and if the choice was correct the chosen move was implemented on the screen. Subjects repeated this training until they satisfied a performance criterion, based on how many points they missed consequent upon choosing less rewarding images. The initial performance criterion allowed four missed points, or less, in a whole training round (out of a maximum of 130 points). This criterion was gradually relaxed, to eight missed points in the second training round, to 12 missed points in the third training round, and to 16 missed points thereafter. Once subjects satisfied the performance criterion without time limit, they repeated the training with only 1500 ms allowed to make each choice, until satisfying the same re-set gradually relaxing criterion. Overall, subjects required an average of 3.4 training rounds (SD 1.0) to learn the initial image-reward associations (1.3 rounds without, and then 2.1 rounds with, a 1500 ms time limit), and 4.3 rounds (SD 1.3) to learn the second set image-reward associations (2.0 rounds without, and 2.3 rounds with, a time limit). Questioning at the end of the experiment validated that subjects had explicit recall for both sets of image-reward associations (mean error 0.36 pts, SEM = 0.07 pts; chance = 4.05 pts).

### Modelling

To test what decision algorithm subjects employed, and in particular, whether they chose moves that had previously been most rewarding from the same starting location (model-free planning), or whether they learned how the state space is structured and used this information to plan ahead (model-based planning), we compared between model-free and model-based algorithms in terms of how well they fitted subjects' actual choices. These models were informed by previous work (*Daw et al., 2005*; *Sutton and Barto, 1998*), adjusted to the present task, and validated using model and parameter recovery tests on simulated data.

### Model-free learning algorithm

Free parameters: $\eta^{\mathrm{MF1}}$, $\eta^{\mathrm{MF2}}$, $\tau^{\mathrm{MF}}$, $\tau'^{\mathrm{MF}}$, $\theta$, $\beta_{1,2}^{\mathrm{MF1}}$, $\beta_2^{\mathrm{MF2}}$, $\gamma_{\mathrm{up,down,left,\,right}}$. This algorithm learns the expected value of performing a given move upon encountering a given image. To do this, the algorithm updates its expectation $Q^{\mathrm{MF}}$ from move $m$ given image $s$ whenever this move is taken and its outcome is observed:

$$Q_{t+1}^{\mathrm{MF1}}\left(s_{t,1}, m_t\right) = Q_t^{\mathrm{MF1}}\left(s_{t,1}, m_t\right) + \eta^{\mathrm{MF1}} \delta_t^{\mathrm{MF1}}, \tag{1}$$

where $s_{t,1}$ is trial $t$'s starting image, $\delta_t^{\mathrm{MF}}$ is the reward prediction error, and $\eta^{\mathrm{MF1}}$ is a fixed learning rate between 0 and 1. Reward prediction errors are computed as the difference between actual and expected outcomes:

$$\delta_t^{\mathrm{MF1}} = R_g\left(s_{t,2}\right) - Q_t^{\mathrm{MF1}}\left(s_{t,1}, m_t\right), \tag{2}$$

where the actual outcome consists of the points associated with the new image to which the move led, $R_g\left(s_{t,2}\right)$. $g = 1$ refers to the initial image-rewards associations, and $g = 2$ refers to the second set of image-rewards associations about which subjects were instructed in the middle of the experiment.

On 2-move trials, the algorithm also learns the expected reward for each pair of moves given each starting image. Thus, another set of Q values is maintained ($Q^{\mathrm{MF2}}$), one for each possible pair of moves for each starting image, and these are updated every time a pair of moves is completed based on the total reward obtained by the two moves. This learning proceeds as described by *Equations 1 and 2*, but with a different learning rate ($\eta^{\mathrm{MF2}}$).

All expected values are initialised to $\theta$, and decay back to this initial value before every update:

$$Q^{\mathrm{MF}} \leftarrow \tau^{\mathrm{MF}} Q^{\mathrm{MF}} + \left(1 - \tau^{\mathrm{MF}}\right)\theta, \tag{3}$$

where $\tau^{\mathrm{MF}}$ determines the degree of value retention. This allows learned expectations to be gradually forgotten.

Following instructed changes to the number of points associated with each image, or to the spatial arrangement of the images, previously learned Q values are of little use. Thus, we allow the Q values to then return back to $\theta$, as in *Equation 3*, but only for a single timestep and with a different, potentially lower, memory parameter $\tau'^{\mathrm{MF}}$.

Finally, the algorithm chooses moves based on a combination of its learned expected values. On 1-move trials, only single-move Q values are considered:

$$\mathrm{p}(m_t = a|s_t) \propto e^{\gamma_m + \beta_1^{\mathrm{MF1}} Q_t^{\mathrm{MF1}}\left(s_{t,1}, m\right)}, \tag{4}$$

where $\gamma_m$ is a fixed bias in favor of move $m$ ($\sum_m \gamma_m = 0$), and $\beta_1^{\mathrm{MF1}}$ is an inverse temperature parameter that weighs the impact of expected values on choice. On 2-move trials, both types of Q values are considered. Thus, the first move is chosen based on a weighted sum of the single-move Q values and the move-pair Q values:

$$\mathrm{p}\left(m_{t,1} = m|s_{t,1}\right) \propto e^{\gamma_m + \beta_2^{\mathrm{MF1}} Q_t^{\mathrm{MF1}}\left(s_{t,1}, m\right) + \beta_2^{\mathrm{MF2}} Q_t^{\mathrm{MF2}}\left(s_{t,1}, m\right)}, \tag{5}$$

wherein the latter are integrated over possible second moves each weighted by its probability:

$$Q_t^{\mathrm{MF2}}\left(s_{t,1}, m\right) = \sum_{m^*} \mathrm{p}\left(m_{t,2} = m^*|s_{t,1}, m_{t,1}\right) Q_t^{\mathrm{MF2}}\left(s_{t,1}, m, m^*\right) \tag{6}$$

Then, in choosing the second move the algorithm takes into account the state to which the first move led:

$$\mathrm{p}\left(m_{t,2} = m|s_{t,1}, m_{t,1}, s_{t,2}\right) \propto e^{\gamma_m + \beta_2^{\mathrm{MF1}} Q_t^{\mathrm{MF1}}\left(s_{t,2}, m\right) + \beta_2^{\mathrm{MF2}} Q_t^{\mathrm{MF2}}\left(s_{t,1}, m_{t,1}, m\right)}. \tag{7}$$

However, when the newly reached image $s_{t,2}$ is not known (i.e., in trials without feedback, or when estimating $\mathrm{p}\left(m_{t,2} = m^*|s_{t,1}, m_{t,1}\right)$ in *Equation 6* before $s_{t,2}$ is reached), $Q^{MF1}$ values are averaged over all settings of $s_{t,2}$.

## Model-based learning algorithm

Free parameters: $\eta^{\mathrm{MB}}$, $\tau^{\mathrm{MB}}$, $\tau'^{\mathrm{MB}}$, $\rho$, $\omega$, $\beta^{\mathrm{MB}}$, $\kappa$, $\gamma_{\mathrm{up,down,left, right}}$. This algorithm learns the probability of transitioning from one image to another following each move. To do this, the algorithm updates its probability estimates, $T$, whenever a move is made and a transition is observed:

$$T_{t+1}\left(s_{t,1}, m_t, s_{t,2}\right) = T_t\left(s_{t,1}, m_t, s_{t,2}\right) + \eta^{\mathrm{MB}} \delta_t^{\mathrm{MB}}, \tag{8}$$

where $\delta_t^{\mathrm{MB}}$ is the image-transition prediction error, and $\eta^{\mathrm{MF}}$ is a fixed learning rate between 0 and 1. Image-transition prediction errors reflect the difference between actual and expected transitions:

$$\delta_t^{\mathrm{MB}} = 1 - T_t\left(s_{t,1}, m_t, s_{t,2}\right). \tag{9}$$

To ensure that transition probabilities sum to 1, the transition matrix is renormalised following every update:

$$\forall s \quad T_{t+1}\left(s_{t,1}, m_t, s\right) \leftarrow \frac{T_{t+1}\left(s_{t,1}, m_t, s\right)}{\sum_{s'} T_{t+1}\left(s_{t,1}, m_t, s'\right)}. \tag{10}$$

Learning may also take place with respect to the opposite transition. For instance, if moving right from image $s_{t,1}$ leads to image $s_{t,2}$, the agent can infer that moving left from image $s_{t,2}$ would lead to image $s_{t,1}$. Such inference is modulated in the algorithm by free parameter $\rho$:

$$T_{t+1}\left(s_{t,2}, \tilde{m}_t, s_{t,1}\right) = T_t\left(s_{t,2}, \tilde{m}_t, s_{t,1}\right) + \rho \eta^{\mathrm{MB}} \delta_t^{'\mathrm{MB}} \tag{11}$$

where $\tilde{m}_t$ is the opposite of $m_t$, and $\delta'$ is the opposite transition prediction error:

$$\delta_t^{'\mathrm{MB}} = 1 - T_t\left(s_{t,2}, \tilde{m}_t, s_{t,1}\right). \tag{12}$$

Self-transitions are impossible and thus their probability is initialised to 0. All other transitions are initialised with uniform probabilities, and these probabilities decay back to their initial values before every update:

$$T \leftarrow \tau^{\mathrm{MB}} T + \left(1 - \tau^{\mathrm{MB}}\right) \frac{1}{7}, \tag{13}$$

where $\tau^{\mathrm{MB}}$ is the model-based memory parameter. A low $\tau^{\mathrm{MB}}$ results in faster decay of expected transition probabilities towards uniform distributions, decreasing the impact of MB knowledge on choice.

When instructed about changes to the image locations, the agent rearranges its transition probabilities based on the instructed changes with limited success, as indexed by free parameter $\omega$:

$$T \leftarrow (1 - \omega)T + \omega T^{\mathrm{rearranged}}. \tag{14}$$

Since some subjects may simply reset their transition matrix following instructed changes, the algorithm also 'forgets' after such instruction, as in *Equation 13*, but only for a single time point and with a different memory parameter, $\tau^{'}\mathrm{MB}$.

Finally, the probability the algorithm will choose a given move when encountering a given image depends on its model-based estimate of the move's expected outcome:

$$\mathrm{p}\left(m_t = m | s_{t,1}\right) \propto e^{\gamma_m + \beta^{\mathrm{MB}} Q_t^{\mathrm{MB}}\left(s_{t,1}, m\right)}. \tag{15}$$

The algorithm estimates expected outcomes by multiplying the number of points associated with an image with the probability of transitioning to that image, integrating over all potential future images:

$$Q_t^{MB}\left(s_{t,1}, m\right) = \sum_s T_t\left(s_{t,1}, m, s\right) R_g(s). \tag{16}$$

When two moves are allowed, the calculation also accounts for the number of points obtainable with the second move, $m_{t,2}$:

$$Q_t^{MB}\left(s_{t,1}, m\right) = \sum_s T_t\left(s_{t,1}, m, s\right) \left(R_g(s) + \kappa \max_{m'} \sum_{s'} T_t\left(s, m', s'\right) R_g\left(s'\right)\right), \tag{17}$$

where $\kappa$ is a fractional parameter that determines the degree to which reward obtained by the second move is taken into account.

Following the first move, *Equation 15* is used to choose a second move based on the observed new location ($s_{t,2}$). However, if the next location is not shown (i.e., in trials without feedback), the

agent chooses its second move by integrating *Equation 15* over the expected $s_{t,2}$, as determined by $T_t(s_{t,1}, m_{t,1}, s_{t,2})$.

## MF-MB hybrid algorithm

This algorithm employs both model-free (MF) and model-based (MB) planning, choosing moves based on a combination of the expected values estimated by the two learning processes. In 1-move trials this is implemented as:

$$\mathrm{p}(m_t = m | s_t) \propto e^{\gamma_m + \beta_1^{\mathrm{MF1}} Q_t^{\mathrm{MF1}}(s_{t,1}, m) + \beta^{\mathrm{MB}} Q^{\mathrm{MB}}(s_{t,1}, m)}, \tag{18}$$

In 2-move trials, the algorithm makes a choice based on a combination of the model-based Q values and both the single-move and two-move model-free Q values. For the first move, the combination is:

$$\mathrm{p}(m_{t,1} = m | s_{t,1}) \propto e^{\gamma_m + \beta_2^{\mathrm{MF1}} Q_t^{\mathrm{MF1}}(s_{t,1}, m) + \beta_2^{\mathrm{MF2}} Q_t^{\mathrm{MF2}}(s_{t,1}, m) + \beta^{\mathrm{MB}} Q^{\mathrm{MB}}(s_{t,1}, m)} \tag{19}$$

with $Q^{\mathrm{MB}}(s_{t,1}, m)$ computed according to *Equation 17*. For the second move, the choice is made according to:

$$\mathrm{p}(m_{t,2} = m | s_{t,2}) \propto e^{\gamma_m + \beta_2^{\mathrm{MF1}} Q_t^{\mathrm{MF1}}(s_{t,2}, m) + \beta_2^{\mathrm{MF2}} Q_t^{\mathrm{MF2}}(s_{t,1}, m_{t,1}, m) + \beta^{\mathrm{MB}} Q^{\mathrm{MB}}(s_{t,2}, m)} \tag{20}$$

When the image is not shown following the first move (i.e., in a no-feedback trial), the agent averages the model-free values over all images.

## Parameter fitting

To fit the free parameters of the different algorithms to subjects' choices, we used an iterative hierarchical expectation-maximisation procedure (*Bishop, 2006*). We first sampled 10000 random settings of the parameters from predefined group-level prior distributions. Then, we computed the likelihood of observing subjects' choices given each setting, and used the computed likelihoods as importance weights to re-fit the parameters of the group-level prior distributions. These steps were repeated iteratively until model evidence ceased to increase (see Model Comparison below for how model evidence was estimated). This procedure was then repeated with 31623 samples per iteration, and finally with 100000 samples per iteration. To derive the best-fitting parameters for each individual subject, we computed a weighted mean of the final batch of parameter settings, in which each setting was weighted by the likelihood it assigned to the subject's choices. Fractional parameters ($\eta^{\mathrm{MF}}$, $\tau^{\mathrm{MF}}$, $\tau'^{MF}$, $\eta^{\mathrm{MB}}$, $\tau^{\mathrm{MB}}$, $\tau'^{\mathrm{MB}}$, $\rho$, $\omega$, $\alpha$) were modelled with Beta distributions (initialised with shape parameters $a = 1$ and $b = 1$) and their values were log-transformed for the purpose of subsequent analysis. Initial Q values ($\theta$) and bias parameters ($\gamma_{\mathrm{up}}$, $\gamma_{\mathrm{down}}$, $\gamma_{\mathrm{left}}$, $\gamma_{\mathrm{right}}$) were modelled with normal distributions (initialised with $\mu = 0$ and $\sigma = 1$) to allow for both positive and negative effects, and all other parameters were modeled with Gamma distributions (initialised with shape = 1, scale = 1).

## Algorithm comparison

We compared between pairs of algorithms, in terms of how well each accounted for subjects' choices, by means of the integrated Bayesian Information Criterion (iBIC; *Eldar et al., 2016*; *Huys et al., 2012*). To do this, we estimated the evidence in favour of each model ($\mathfrak{L}$) as the mean likelihood of the model given 100000 random parameter settings drawn from the fitted group-level priors. We then computed the iBIC by penalising the model evidence to account for algorithm complexity as follows: $\mathrm{iBIC} = -2 \ln \mathfrak{L} + k \ln n$, where $k$ is the number of fitted parameters and $n$ is the number of subject choices used to compute the likelihood. Lower iBIC values indicate a more parsimonious fit.

## Algorithm and parameter recovery tests

We tested whether our dataset was sufficiently informative to distinguish between the MF, MB and hybrid algorithms and recover the correct parameter values. For this purpose, we generated 10 simulated datasets using each algorithm and applied our fitting and comparison procedures to each

dataset. To reduce processing time, only 10000 parameter settings were sampled. To maximise the chances of confusion between algorithms, we implemented all algorithms with the parameter values that best fitted subjects' choices. Algorithm comparison implicated the correct algorithm in each of the 30 simulated datasets, and the parameters values that best fitted the simulated data consistently correlated with the actual parameter values used to generate these data (Pearson's $r$: $M = 0.57$, $SEM = 0.05$). This correlation was stronger for parameters whose values were used for multiple trials when computing the fit to data (e.g., learning rates and inverse temperature parameters; $M = 0.67$, $SEM = 0.04$).

## Additional algorithms

To test whether the algorithms described above were suitable for describing subjects' behaviour, we compared them to several additional algorithms, all of which failed to fit subjects' choices as well as the above counterparts, and so we do not describe them in detail. These alternative algorithms included a MF algorithm that only learns single-move Q values, but employs temporal difference learning (*O'Doherty et al., 2003*) to backpropagate second outcomes in 2-move trials back to the Q values of the starting location (BIC = 41559); a MB algorithm that employs Bayesian inference with a uniform Dirichlet prior (*Bishop, 2006*) (whose concentration parameter was set to 1) to learn the multinomial distributions that compose the state transition matrix (BIC = 43301); a MF-MB hybrid algorithm where state-transition expectations are only used to account for prospective second-move Q values when choosing the first move in 2-move trials (BIC = 40920); and an algorithm that combines two MF algorithms with different parameters (BIC = 40715).

## MEG acquisition

MEG was recorded continuously at 600 samples/second using a whole-head 275-channel axial gradiometer system (CTF Omega, VSM MedTech, Canada), while subjects sat upright inside the scanner. A projector displayed the task on a screen ~80 cm in front of the subject. Subjects made responses by pressing a button box, using their left hand for 'left' and 'up' choices and their right hand for 'right' and 'down' choices. Pupil size and eye gaze were recorded at 250 Hz using a desktop-mounted EyeLink II eyetracker (SR Research).

## MEG preprocessing

Preprocessing was performed using the Fieldtrip toolbox (*Oostenveld et al., 2011* in MATLAB (MathWorks). Data from two sensors were not recorded due to a high level of noise detected in routine testing. Data were first manually inspected for jump artefacts. Then, independent component analysis was used to remove components that corresponded to eye blinks, eye movement and heart beats. Based on previous experience (*Eldar et al., 2018*), we expected stimuli to be represented in low frequency fluctuations of the MEG signal. Therefore, to remove fast muscle artefacts and slow movement artefacts, we low-pass filtered the data with a 20 Hz cutoff frequency using a sixth-order Butterworth IIR filter, and we baseline-corrected each trial's data by subtracting the mean signal recorded during the 400 ms preceding trial onset. Trials in which the average standard deviation of the signal across channels was at least 3 times greater than median were excluded from analysis (0.4% of trials, SEM 0.2%). Finally, the data were resampled from 600 Hz to 100 Hz to conserve processing time and improve signal to noise ratio. Therefore, data samples used for analysis were length 273 vectors spaced every 10 ms.

## Pre-task stimulus exposure

To allow decoding of images from MEG we instructed subjects to identify each of the images in turn (*Figure 2—figure supplement 1a*). On each trial, the target image was indicated textually (e.g., 'FACE') and then an image appeared on the screen. Subjects' task was to report whether the image matched (LEFT button) or did not match (RIGHT button) the preceding text. 20% of presented images did not match the text. The task continued until subjects correctly identified each of the images at least 25 times. Subjects were highly accurate on both match (M = 97.2%, SEM = 0.4%) and no-match (M = 90.2%, SEM = 0.6%) trials. To ensure robust decoding from MEG, we chose eight images that differed in colour, shape, texture and semantic category (*Isik et al., 2014*; *Carlson et al., 2013*; *Figure 1a*). Importantly, at this point subjects had no knowledge as to what

the main task would involve, nor that the images would be associated with state-space locations and rewards. This ensured that no task information could be represented in the MEG data at this stage.

## MEG decoding

We used support vector machines (SVMs) to decode images and moves from MEG. All decoders were trained on MEG data recorded outside of the main state-space task and validated within the task. As in previous work (*Eldar et al., 2018*), we trained a separate decoder for each time bin between 150 and 600 ms following the relevant event, either image onset or move choice, resulting in 46 decoders whose output was averaged. Averaging over decoders trained at different time points reduces peak decodability following stimulus onset, but can increase decodability of stimuli that are being processed when not on the screen (*Eldar et al., 2018*). To avoid over-fitting, training and testing were performed on separate sets of trials following a 5-fold cross validation scheme. These analyses were performed using LIBSVM's implementation of the C-SVC algorithm with radial basis functions (*Chang and Lin, 2011*). Decoder training and testing were performed with each of 16 combinations of the algorithms' cost parameter ($10^{-1}$, $10^{0}$, $10^{1}$, $10^{2}$) and basis-function concentration parameter ($10^{-2}/n$, $10^{-1}/n$, $10^{0}/n$, $10^{1}/n$), where $n$ is the number of MEG features (273 channels). Where classes differed in number of instances, weighting was used to ensure classes were equally weighted.

To decode the probability of each of eight possible images being presented (8-way classification), we used MEG data recorded during pre-task stimulus exposure. Decoding was evaluated based on the mean probability the decoders assigned to the presented image. To decode the probability of each of the four possible moves (LEFT, RIGHT, UP, DOWN) being chosen (4-way classification), we used MEG data recorded during the image-reward training. For both types of decoder, the parameter combination of cost = $10^{2}$ and concentration = $10^{-2}/n$ yielded the best cross-validated decoding performance and was thus used for all ensuing analyses, wherein decoders trained on pre-task stimulus exposure data were applied to main task data.

## Sequenceness measure

To investigate how representations of different images related to one another in time, we used a measure recently developed for detecting sequences of representations in MEG; (*Kurth-Nelson et al., 2016*). 'Sequenceness' is computed as the difference between the cross-correlation of two images' decodability time-series with positive and negative time lags. By relying on asymmetries in the cross-correlation function, this measure detects sequential relationships even between closely correlated (or anti-correlated) time series, as we have previously demonstrated on simulated time series (*Eldar et al., 2018*). Positive values indicate that changes in the first time series are followed by similar changes in the second time series ('forward sequenceness'), negative values indicate the reverse sequence ('backward sequenceness'), and zero indicates no sequential relationship. As in previous work, cross correlations were computed between the z-scored time series over 400 ms sliding windows with time lags of up to 200 ms. This timescale is sufficient for capturing the relationship between successive alpha cycles, which is important given the possibility that such oscillations may reflect temporal quanta of information processing (*Busch and VanRullen, 2014*).

## Bayesian hierarchical Gaussian process time series analysis

To determine whether sequenceness time-series recorded following outcomes provided robust evidence of replay that correlated with individual index of behavioural flexibility, we modelled mean sequenceness time-series as Gaussian Processes with squared exponential kernels. Such Gaussian Processes explicitly account for dependencies between timepoints within a time series, as a function of the lengths of time between those points. A group-level Gaussian Process captures the time series' systematic deviations from zeros, either on average or as a function of two predictors: subject IF and surprise about the outcome. The deviations of each individual time series from the predictions made by this group-level process can themselves exhibit dependencies between timepoints as a function of distance. Therefore, we accounted for these deviations by means of another set of Gaussian Processes, one for each modelled time series, which were added to the predictions of the group-level process.

In theory, this model can be fit using MCMC sampling to all trial-by-trial sequenceness time series. However, fitting the model to such an amount of data proved infeasible within a reasonable timeframe. Thus, we reduced the data to four mean sequenceness time series per subject: sequenceness encoding the last (as in *Figure 3a*) or penultimate (as in *Figure 3b*) transition following highly or weakly surprising outcomes. High and low surprise were determined based on the state prediction error generated by the hybrid algorithm, whose parameters were fitted to the individual subject's choices (i.e., high – above-mean prediction error, low – below-mean prediction error). Since we assumed last and penultimate transitions could be replayed at different timepoints, these two types of time series each had their own group-level Gaussian Process. To account for the factors of IF and surprise, for each time series, the group-level process was multiplied by a weighted linear combination of the two factors, their interaction, and an intercept (thus involving four parameters: $\beta$, $\beta^{\text{subject}}$, $\beta^{\text{surprise}}$, $\beta^{\text{interaction}}$). The group- and individual-level Gaussian Processes were parameterised by different length-scales ($\rho^{\text{group}}$, $\rho^{\text{inidivdual}}$) and marginal standard deviations ($\alpha^{\text{group}}$, $\alpha^{\text{individual}}$), and a standard deviation parameter ($\sigma$) accounted for additional normally distributed noise across all observations.

Bayesian estimation was performed in R (*R Development Core Team, 2018*) using the STAN (*Carpenter et al., 2017*) package for Markov Chain Monte Carlo (MCMC) sampling. All predictor variables were standardised. All prior distributions were set so as to be weakly informative and have broad range on the scale of the variables; (*Kruschke, 2014*). This included the following: $\beta$ coefficients were drawn from normal distributions with a mean of zero and a standard deviation of 10; Standard deviations parameters ($\alpha$, $\sigma$) were drawn from a normal distribution truncated to positive values, with a mean of zero and a standard deviation that matches the standard deviation of the predicted variable; Length-scales ($\rho$) were drawn from log-normal distributions whose mean is the geometric mean of two extremes: the distance in time between two successive timepoints, and the distance in time between the first and last timepoints; Half of the difference between these two values was used as the standard deviation of the priors. For the sake of identifiability, $\beta^{\text{interaction}}$ was limited to positive values. Note this does not limit the model to positive interactions, since all coefficient are multiplied by the group-level Gaussian Processes, which might be negative.

We ran six MCMC chains each for 1400 iterations, with the initial 400 samples used for warmup. STAN's default settings were used for all other settings. Examining the results showed there were no divergent transitions, and all parameters were estimated with effective sample sizes larger than 1000 and shrink factors smaller than 1.1. Posterior predictive checks showed good correspondence between the real and generated data (*Figure 1c*; *Figure 1—figure supplement 4*).

## Decodability time series analyses

Decodability was tested for difference from zero and covariance with individual flexibility using the Bayesian Gaussian Process approach outlined above with the exclusion of the surprise predictor, which is inapplicable to timepoints that precede outcome onset.

## Other statistical methods

Significance tests were conducted using nonparameteric methods that do not assume specific distributions. Differences from zero were tested using 10000 samples of bias-corrected and accelerated Bootstrap with default MATLAB settings. Correlations and differences between groups were tested by comparison to null distributions generated by 10000 permutations of the pairing between the two variables of interest. All tests are two-tailed.

## Acknowledgements

We thank Zeb Kurth-Nelson and Yunzhe Liu for helpful comments on a previous version of the manuscript. EE holds an Alon Fellowship from the Israeli Council for Higher Education. PD is funded by the the Max Planck Society and the Humboldt Foundation. RJD holds a Wellcome Trust Investigator award (098362/Z/12/Z). The Max Planck UCL Centre for Computational Psychiatry and Ageing Research is a joint initiative supported by the Max Planck Society and University College London. The Wellcome Centre for Human Neuroimaging is supported by core funding from the Wellcome Trust (091593/Z/10/Z).

## Additional information

### Funding

| Funder | Grant reference number | Author |
|---|---|---|
| Council for Higher Education | Alon Fellowship | Eran Eldar |
| Max Planck Society | | Peter Dayan |
| Alexander von Humboldt Foundation | | Peter Dayan |
| Wellcome Trust | Wellcome Trust Investigator award (098362/Z/12/Z) | Raymond J Dolan |
| Max Planck Society | | Raymond J Dolan |

The funders had no role in study design, data collection and interpretation, or the decision to submit the work for publication.

### Author contributions

Eran Eldar, Conceptualization, Data curation, Software, Formal analysis, Supervision, Validation, Investigation, Visualization, Methodology, Writing - original draft, Project administration, Writing - review and editing; Gaëlle Lièvre, Data curation, Investigation, Methodology; Peter Dayan, Formal analysis, Supervision, Methodology, Writing - review and editing; Raymond J Dolan, Resources, Supervision, Funding acquisition, Project administration, Writing - review and editing

### Author ORCIDs

Eran Eldar (iD) https://orcid.org/0000-0001-8988-6124
Gaëlle Lièvre (iD) https://orcid.org/0000-0002-6914-1894
Peter Dayan (iD) https://orcid.org/0000-0003-3476-1839
Raymond J Dolan (iD) https://orcid.org/0000-0001-9356-761X

### Ethics

Human subjects: The experimental protocol was approved by the UCL Research Ethics Committee, under Project ID Number 9929/002, and informed consent was obtained from all subjects.

### Decision letter and Author response

Decision letter https://doi.org/10.7554/eLife.56911.sa1
Author response https://doi.org/10.7554/eLife.56911.sa2

## Additional files

### Supplementary files

• Transparent reporting form

### Data availability

The data and the custom code have been deposited in the Open Science Framework under https: https://doi.org/10.17605/OSF.IO/GUHJE.

The following dataset was generated:

| Author(s) | Year | Dataset title | Dataset URL | Database and Identifier |
|---|---|---|---|---|
| Eldar E | 2020 | The roles of online and offline replay | https://osf.io/guhje/ | Open Science Framework, 10.17605/OSF.IO/GUHJE |

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
