## [Decision Letter]

**Acceptance summary:**

The findings presented in this paper describe the relationship between cognitive flexibility and replay in humans. A major strength of this study is that it shows a direct relationship between replay and individual differences in behavior. This is a major step toward a better understanding of how replay contributes to reinforcement learning and planning.

**Decision letter after peer review:**

Thank you for submitting your article "The roles of online and offline replay in planning" for consideration by *eLife*. Your article has been reviewed by three peer reviewers, and the evaluation has been overseen by by a Reviewing Editor and Kate Wassum as the Senior Editor. The following individual involved in review of your submission has agreed to reveal their identity: Samuel J Gershman (Reviewer #2).

The reviewers have discussed the reviews with one another and the Reviewing Editor has drafted this decision to help you prepare a revised submission.

As the editors have judged that your manuscript is of interest, but as described below that additional experiments or analyses are required before it is published, we would like to draw your attention to changes in our revision policy that we have made in response to COVID-19 (https://elifesciences.org/articles/57162). First, because many researchers have temporarily lost access to the labs, we will give authors as much time as they need to submit revised manuscripts. We are also offering, if you choose, to post the manuscript to bioRxiv (if it is not already there) along with this decision letter and a formal designation that the manuscript is 'in revision at *eLife*'. Please let us know if you would like to pursue this option. (If your work is more suitable for medRxiv, you will need to post the preprint yourself, as the mechanisms for us to do so are still in development.)

Summary:

This paper presents evidence for the relationship between cognitive flexibility and replay decoded from MEG signals. The work significantly extends prior results, by connecting replay with individual differences in behavior. All reviewers agreed that this work is technically solid and the experimental results compelling, distinguishing between forward and backward replay as well as online vs. offline replay. Reviewers also found the paper clearly written but somewhat dense. Reviewers also noted some issues that should be addressed with additional analyses and/or data before the manuscript can be accepted for publication.

Essential revisions:

1) Evidence for replay is based on decoding of the previous state at the time of outcome. How much of this decoding can be explained by sensory input or sensory processing of the previous image, rather than replay as a separate neural event? Figure 2A shows that decoding might still be above chance after image offset. And Figure 2—figure supplement 2B suggests that decoding of the previous state is already above chance at and likely before outcome onset. Can the authors show that decoding is indeed driven by a replay-like mechanisms rather than sensory/perceptual processing of the previous state? For instance, how does decoding evolve in the second(s) before outcome onset? If decoding is driven by replay at outcome onset, should we not expect decoding to peak after outcome onset? Also decoding at outcome onset should not depend on the length of the ISI between the offset of the previous image and outcome onset. (Alternatively, if decoding is driven by sensory processing, decodability should be higher for shorter compared to longer ISIs.) Finally, the authors could use data from a more passive viewing task to get an idea of how long decoding should be expected to remain above chance after image offset in the absence of replay. To support the main conclusions about replay, it would be important to show that decoding is not simply driven by residual processing of the previous image.

2) Where in the brain does replay occur? It would be interesting to see which of the 273 MEG channels contribute to successful decoding. If the authors could show that putative replays is localized near hippocampal sensors rather than, e.g., visual ones, this may also help to convince that decoding of the previous state is not simply based on visual processing (see essential revision point 1).

3) Some reviewers found the manuscript very hard to read. Results are consistently stated in terms of technical constructs and/or methods that require substantial acquaintance with the authors' previous work. For example, the Introduction cites a copious literature on rodent replay and preplay in hippocampus without justifying the premise that this can be successfully decoded from MEG in humans. Another example: subsection “Bayesian hierarchical Gaussian Process time series analysis”, second paragraph. This was considered far too dense, particularly since several critical modeling choices were not well justified. The concern is that this style forces non-specialist readers (and even some specialists) to do a lot of work that might be avoided with more motivation and explication. A better description of some of the key analysis methods (the sequenceness analysis) in the main text would be helpful. In this regard, it might also help to have some kind of conceptual figure relating individual flexibility, model-free vs. model-based learners, preplay/replay, on-task/off-task, and other relevant constructs so readers can see their relationships. The authors present a number of different results in different epochs pertaining to different claims, but it is hard to see the coherent whole, particularly for those not already heavily invested in these debates.

4) The authors invoke the recent Mattar and Daw theory of hippocampal replay, arguing that it predicts that "subjects should be more disposed to replay trajectories that they might not want to choose again, rather than trajectories whose choice reflects a firm policy". We gather that this is related to Figure 5 in Mattar and Daw. But aren't the theory's predictions more complicated? Mattar and Daw also point out that enhanced replay should occur when an agent is likely to *repeat* an action, for example when receiving a larger than expected reward. Another issue is that Mattar and Daw's theory specifically predicts an effect on reverse replay ("backward sequenceness" here) not forward replay ("forward sequenceness"), but the authors report an effect for forward sequenceness.

5) "off-task sequenceness negatively correlated with IF (Figure 3). This association of sequenceness during rest with low flexibility is consistent with a proposed role of offline replay in establishing model-free policies." The logic of these predictions is understandable based on algorithms like Dyna, but couldn't this also go in the opposite direction? For example, if one thinks about the Gershman, Markman and Otto, 2014 studies, their revaluation index is conceptually similar to the IF measure used here. But in that case large revaluation (i.e., greater flexibility) was used to argue in favor of more replay in a Dyna-like manner, and this hypothesis received more direct confirmation from the Momennejad *eLife* paper. How can we reconcile these two interpretations of flexibility (as reflecting more vs. less replay)?

---

## [Author Response]

Essential revisions:1) Evidence for replay is based on decoding of the previous state at the time of outcome. How much of this decoding can be explained by sensory input or sensory processing of the previous image, rather than replay as a separate neural event? Figure 2A shows that decoding might still be above chance after image offset. And Figure 2—figure supplement 2B suggests that decoding of the previous state is already above chance at and likely before outcome onset. Can the authors show that decoding is indeed driven by a replay-like mechanisms rather than sensory/perceptual processing of the previous state? For instance, how does decoding evolve in the second(s) before outcome onset? If decoding is driven by replay at outcome onset, should we not expect decoding to peak after outcome onset? Also decoding at outcome onset should not depend on the length of the ISI between the offset of the previous image and outcome onset. (Alternatively, if decoding is driven by sensory processing, decodability should be higher for shorter compared to longer ISIs.) Finally, the authors could use data from a more passive viewing task to get an idea of how long decoding should be expected to remain above chance after image offset in the absence of replay. To support the main conclusions about replay, it would be important to show that decoding is not simply driven by residual processing of the previous image.

The reviewers raise an important question about the interpretation of previous-state decoding on which evidence of replay is based. We thank the reviewers for their helpful suggestions as to how to address this question. In a new analysis, we examine decodability of each state time-locked to its own offset. The results show that outcomes were indeed followed by increased decodability that correlated with evidence of post-outcome sequenceness (*n* = 40, *r* = 0.32, *p* = 0.04, permutation test). Furthermore, the experiment provides us with a natural comparison between non-terminal states that were followed by outcomes (blue in Figure 2—figure supplement 2C; those from which subjects moved) and terminal states that were not (grey; those that marked the end of a trial). Following the terminal states, we could decode the preceding non-terminal states (the black bar). However, crucially, we could not significantly decode terminal states this long after their offset (difference between non-terminal and terminal states: *p* = 0.03, Permutation test). These results indicate that decoded state representations did not only reflect sensory processing, and were well suited to contribute to post-outcome planning. We now report this new result in Figure 2—figure supplement 2C and note it in the main text:

“As would be expected, the finding of sequenceness was associated with enhanced decoding of previously visited states during the post-outcome epoch (Figure 2—figure supplement 2C).”

2) Where in the brain does replay occur? It would be interesting to see which of the 273 MEG channels contribute to successful decoding. If the authors could show that putative replays is localized near hippocampal sensors rather than, e.g., visual ones, this may also help to convince that decoding of the previous state is not simply based on visual processing (see essential revision point 1).

Understanding where in the brain decoded representations lie is indeed important. However, such analysis has limited practical usefulness for the present study. First, because localizing MEG activity to specific brain areas is highly imprecise in the absence of a proper MRI-based head model (unfortunately subjects were not scanned using MRI). Even with precise head models, identifying hippocampal activity in MEG data is particularly challenging (Meyer et al., 2017). Second, hippocampal replay has been shown at least in some settings to be associated with coherent replay in sensory cortex (Ji and Wilson, 2007). Thus, a differentiation between hippocampal and cortical replay in our data might be unreliable and not necessarily expected.

Nevertheless, delineating the contributions of each sensor to decoder is useful for future reference. We now provide this information in a new sensor map (Figure 2—figure supplement 1B). Our support vector machine decoders do not provide easily interpretable parameters, and thus, we quantified a sensor’s contribution by measuring the correlation between its MEG signal and decoder output, across all main task timepoints. The results indicate all sensors consistently contributed to decoding (most contributing channel: M = 0.269, SE = 0.009; least contributing channel: M = 0.156, SE = 0.004), with a small advantage for posterior sensors. The map has been added to the manuscript as Figure 2—figure supplement 1B and is referred to in the main text.

3) Some reviewers found the manuscript very hard to read. Results are consistently stated in terms of technical constructs and/or methods that require substantial acquaintance with the authors' previous work. For example, the Introduction cites a copious literature on rodent replay and preplay in hippocampus without justifying the premise that this can be successfully decoded from MEG in humans. Another example: subsection “Bayesian hierarchical Gaussian Process time series analysis”, second paragraph. This was considered far too dense, particularly since several critical modeling choices were not well justified. The concern is that this style forces non-specialist readers (and even some specialists) to do a lot of work that might be avoided with more motivation and explication. A better description of some of the key analysis methods (the sequenceness analysis) in the main text would be helpful. In this regard, it might also help to have some kind of conceptual figure relating individual flexibility, model-free vs. model-based learners, preplay/replay, on-task/off-task, and other relevant constructs so readers can see their relationships. The authors present a number of different results in different epochs pertaining to different claims, but it is hard to see the coherent whole, particularly for those not already heavily invested in these debates.

We apologise that the original version was hard for readers. We thank the reviewers for helpful suggestions as to how to make the paper more readable. We have substantially revised the manuscript in accordance with these suggestions, including adding the following changes:

– We now devote a paragraph in the Introduction to highlight how recent advances make it possible to detect evidence of replay in humans using MEG):

“Despite the wide-ranging behavioural implications of a distinction between model-based and model-free planning (Kurdi, Gershman and Banaji, 2019; Crockett, 2013; Everitt and Robbins, 2005; Gillan, Fineberg and Robbins, 2017), and much theorising on the role of replay in one or the other form of planning, to date there is little data indicating whether online and offline replay have complementary or contrasting impacts in this regard. […] Thus, finally, the relationships between elements’ representational probability time series are examined to determine whether pairs of elements tended to be represented sequentially, one after the other.”

– We explain the ‘sequenceness’ measure in a new schematic figure (Figure 2B).

– We clarify the details of, and motivation for, the different design choices made in the hierarchical Gaussian Process analysis. We reproduce below a key excerpt added to that effect:

“To determine whether sequenceness time-series recorded following outcomes provided robust evidence of replay that correlated with individual index of behavioural flexibility, we modelled mean sequenceness time-series as Gaussian Processes with squared exponential kernels. […] Therefore, we accounted for these deviations by means of another set of Gaussian Processes, one for each modelled time series, which were added to the predictions of the group-level Process.”

–We complement the discussion with a schematic figure that provides a coherent view of the main findings (Figure 5).

4) The authors invoke the recent Mattar and Daw theory of hippocampal replay, arguing that it predicts that "subjects should be more disposed to replay trajectories that they might not want to choose again, rather than trajectories whose choice reflects a firm policy". We gather that this is related to Figure 5 in Mattar and Daw. But aren't the theory's predictions more complicated? Mattar and Daw also point out that enhanced replay should occur when an agent is likely to repeat an action, for example when receiving a larger than expected reward. Another issue is that Mattar and Daw's theory specifically predicts an effect on reverse replay ("backward sequenceness" here) not forward replay ("forward sequenceness"), but the authors report an effect for forward sequenceness.

Mattar and Daw’s perspective indeed predicts enhanced replay also when an agent becomes more likely to repeat an action. This prediction holds only if an outcome is surprising, such that the information it provides is not already embedded in the agent’s policy. In our experiment, the most natural implication is that replay should be enhanced following surprising outcomes, and consistent with their perspective, our results show that more surprising outcomes were in fact followed by stronger sequenceness (subsection “On-task replay is induced by prediction errors and associated with high flexibility”, last paragraph). As noted by the reviewers, one critical difference from Daw and Mattar here is that we primarily find forward sequenceness following outcomes, where Daw and Mattar predict backward sequenceness. We now highlight this difference explicitly in the Discussion:

“Two main aspects of our results deviate from this normative perspective. First, we primarily find forward sequenceness following outcomes.”

That said, what we see as the most crucial element in Daw and Mattar (as well as in the work on retrospective revaluation; Gershman et al., 2014; Momennejad et al., 2018) is that replay contributes not only to knowledge about the structure of the state space, but also to planning (e.g., by updating action values). An association with prediction errors is not sufficient to substantiate this claim. Prediction errors are of course coupled with policy updates in our model of the task, but that is an assumption built into the model. Thus, what is needed is to examine whether sequenceness is associated with concrete evidence of policy update.

A simple test for this prediction is provided by the fact that subjects’ policy updates at the beginning of each experimental phase are more likely to give rise to behavioral change than to repetition. This is because subjects begin each phase with partial knowledge acquired in the previous phase (as evidenced by above chance performance; Figure 1C), and then go on to improve their policies further as they gain additional experience. Therefore, at this point, some of a subject’s choices already reflect a well-informed policy, and such choices are less likely to induce policy updates and are more likely to be repeated. By contrast, subjects’ more poorly informed choices are more likely to induce policy updates and are less likely to be repeated. Such variance among choices exists in many bandit tasks, but it is likely to be especially pronounced in the present experiment since outcomes are deterministic but numerous, such that learning and forgetting are fast (the average modelled MB forgetting rate was 0.3). Thus, examining behavioral change provides a convenient marker of policy update that is not dependent on specific modeling assumptions. We now clarify this logic more explicitly in the main text:

“Recent theorising regarding the role of replay in planning argues that replay is preferentially induced when there is benefit to updating one’s policy (Mattar and Daw, 2018). […] Thus, a role for replay in planning predicts that subjects should be more disposed to replay trajectories that they might not want to choose again, rather than trajectories whose choice reflects a firm policy.”

5) "off-task sequenceness negatively correlated with IF (Figure 3). This association of sequenceness during rest with low flexibility is consistent with a proposed role of offline replay in establishing model-free policies." The logic of these predictions is understandable based on algorithms like Dyna, but couldn't this also go in the opposite direction? For example, if one thinks about the Gershman, Markman and Otto, 2014 studies, their revaluation index is conceptually similar to the IF measure used here. But in that case large revaluation (i.e., greater flexibility) was used to argue in favor of more replay in a Dyna-like manner, and this hypothesis received more direct confirmation from the Momennejad eLife paper. How can we reconcile these two interpretations of flexibility (as reflecting more vs. less replay)?

The reviewers raise an interesting question regarding the relationship between the present findings and previous findings on retrospective revaluation. Whereas previous findings indicate offline replay is associated with behavioral change, our findings suggest it is associated with decreased flexibility. We consider the two lines of work can be reconciled by considering the scale of flexibility each work emphasizes. While retrospective revaluation manifests in the comparison between learning and test phases separated by additional phases several minutes long, our experiment requires subjects implement different stimulus-response mappings in contiguous trials. An algorithm like Dyna would only be able to afford such flexibility if it can furnish multiple stimulus-response mappings that can be summoned by context. It is possible such flexibility goes beyond the capabilities of forms of Dyna that might be implemented in the brain. We now discuss this issue in relation to previous work in the main text:

“This aspect of the task might explain an apparent discrepancy with previous work on retrospective revaluation (Momennejad et al., 2018; Gershman, Markman and Otto, 2014), which indicates offline replay is associated with a greater degree of behavioural change. […] It is possible such flexibility goes beyond the capabilities of any forms of DYNA that might be implemented in the brain.”